# Hexane Insoluble Fraction from Purple Rice Extract Retards Carcinogenesis and Castration-Resistant Cancer Growth of Prostate through Suppression of Androgen Receptor Mediated Cell Proliferation and Metabolism

**DOI:** 10.3390/nu12020558

**Published:** 2020-02-20

**Authors:** Ranchana Yeewa, Aya Naiki-Ito, Taku Naiki, Hiroyuki Kato, Shugo Suzuki, Teera Chewonarin, Satoru Takahashi

**Affiliations:** 1Department of Experimental Pathology and Tumor Biology, Nagoya City University Graduate School of Medical Sciences, 1-Kawasumi, Mizuho-cho, Mizuho-ku, Nagoya 467-8601, Japan; yeewa.ranchana@gmail.com (R.Y.); naiki@med.nagoya-cu.ac.jp (T.N.); h.kato@med.nagoya-cu.ac.jp (H.K.); suzuki.shugo@med.osaka-cu.ac.jp (S.S.); sattak@med.nagoya-cu.ac.jp (S.T.); 2Department of Biochemistry, Faculty of Medicine, Chiang Mai University, 110 Intravaroros Rd., Sripoom, Muang, Chiang Mai 50200, Thailand

**Keywords:** prostate cancer, castration-resistant prostate cancer, purple rice extract, androgen receptor

## Abstract

Prostate cancer and castration-resistant prostate cancer (CRPC) remain major health challenges in men. In this study, the inhibitory effects of a hexane insoluble fraction from a purple rice ethanolic extract (PRE-HIF) on prostate carcinogenesis and CRPC were investigated both *in vivo* and *in vitro*. In the Transgenic Rat for Adenocarcinoma of Prostate (TRAP) model, 1% PRE-HIF mixed diet-fed rats showed a significantly higher percentage of low-grade prostatic intraepithelial neoplasia and obvious reduction in the incidence of adenocarcinoma in the lateral lobes of the prostate. Additionally, 1% PRE-HIF supplied diet significantly suppressed the tumor growth in a rat CRPC xenograft model of PCai1 cells. In LNCaP and PCai1 cells, PRE-HIF treatment suppressed cell proliferation and induced G0/G1 cell-cycle arrest. Furthermore, androgen receptor (AR), cyclin D1, cdk4, and fatty acid synthase expression were down-regulated while attenuation of p38 mitogen-activated protein kinase, and AMP-activated protein kinase α activation occurred in PRE-HIF treated prostate cancer cells, rat prostate tissues, and CRPC tumors. Due to consistent results with PRE-HIF in PCai1 cells, cyanidin-3-glucoside was characterized as the active compound. Altogether, we surmise that PRE-HIF blocks the development of prostate cancer and CRPC through the inhibition of cell proliferation and metabolic pathways.

## 1. Introduction

Prostate cancer is the second most common cancer in the older male population [1]. The development of prostate cancer at an early-stage depends on the activity of androgens and explains why androgen deprivation therapy (ADT) is the primary treatment for patients with this cancer [2]. Unfortunately, most prostate cancers eventually become refractory to ADT and progress to castration-resistant prostate cancer (CRPC), which is incurable and carries a worse prognosis [3]. Accordingly, the prevention of prostate cancer is a goal.

Purple rice (*Oryza sativa* L. *indica*) is a natural product with several health benefits [4,5]. Bioactive compounds of purple rice, especially anthocyanins, possess various biological properties including antioxidant, anti-atherosclerotic, anti-inflammatory, as well as anti-tumor activities [6,7]. Notably, the suppressive effect of crude ethanolic purple rice extract (PRE) on testosterone-induced benign prostatic hyperplasia (BPH) in rats has been reported [8]. In our recent study, we separated PRE into a hexane soluble fraction (PRE-HSF) and a hexane insoluble fraction (PRE-HIF) [9]. PRE-HIF contained anthocyanins, including cyanidin-3-glucoside (C3G) and peonidin-3-glucoside (P3G). On the other hand, PRE-HSF contained other antioxidant metabolites, such as vitamin D derivatives and γ-oryzanol but no anthocyanins. Furthermore, the anthocyanin-rich fraction, PRE-HIF, was identified as the highly active and non-toxic fraction of purple rice extract on the BPH setting [9]. Similar to BPH, the progression of prostate cancer is initially associated with an androgen receptor (AR) signaling pathway. Therefore, we are interested in the purple rice anthocyanins as candidates of prostatic chemopreventive agents.

To investigate a possible protective action of PRE-HIF against prostate cancer, a Transgenic Rat for Adenocarcinoma of Prostate (TRAP) model was used. This model was developed by linking an SV40 early region to a prostate-specific probasin promotor, resulting in the expression of large T antigen oncoprotein that functionally inactivated the tumor suppressors, p53 and Rb, in prostatic epithelium [10]. At 15 weeks of age, non-invasive adenocarcinomas developed within prostates under hormone-dependent conditions [11]. Consequently, the TRAP model has been widely used for exploring the chemopreventive effect of several compounds against prostate carcinogenesis [12,13,14,15,16,17]. In addition to an androgen-sensitive phase, we evaluated the effect of PRE-HIF on androgen-insensitive prostate cancer using PCai1 cells. A novel rat CRPC cell line, PCai1, was previously established from an androgen-nonresponsive tumor [18]. This, therefore, suggests that PCai1 cells may be a good model that illustrates human CRPC [19].

In this study, we firstly describe the protective effects of an anthocyanin-rich purple rice extract against prostate carcinogenesis and CRPC growth. This observation provides the rationale for the use of purple rice as a dietary supplement in the prevention or treatment of prostate cancer.

## 2. Materials and Methods

### 2.1. Extraction of Anthocyanin-Rich Fraction from Purple Rice

A glutinous purple rice (*Oryza sativa* L. *indica*) variety, Kum Doi Saket, was purchased from the Purple Rice Research Unit, Chiang Mai University, Thailand and then was extracted, according to the previously-published method [8]. Briefly, 1 kg of purple rice grains were blended and stirred in 5 L 80% ethanol overnight (ratio 1:5). After filtrated through Whatman filter paper no.1 and evaporated at 40 °C, the concentrated fraction was partially partitioned with an equal volume of hexane, generating hexane soluble (PRE-HSF) and insoluble (PRE-HIF) fractions. Then, both fractions were evaporated, lyophilized, and subsequently kept in a freezer (‒20 °C) until use. The yields of PRE-HIF were 2.16 g per 100 g purple rice grain. Anthocyanins, including C3G and P3G in PRE-HIF, reflected 4.87 ± 0.05 and 2.25 ± 0.02 mg/g extract, respectively [9]. For consistency among batches, lyophilized samples were subjected to high-performance liquid chromatography for phytochemical analysis as described previously [20].

### 2.2. Chemicals and Cell Lines

C3G chloride and P3G chloride were purchased from Tokiwa Phytochemical (Chiba, Japan) and bicalutamide (Casodex^®^; AstraZeneca, Wilmington, USA). The human prostate cancer cell line, LNCaP, from the American Type Culture Collection (Manassas, VA, USA), was cultured in RPMI (Thermo Fisher Scientific, MA, USA) with 10% fetal bovine serum (FBS), while the original rat CRPC cell line, PCai1, was grown in 10% FBS supplemented DMEM (Thermo Fisher Scientific). Cell cultures were maintained in a humidified incubator with 5% CO_2_/95% air at 37 °C.

### 2.3. Animals

Male heterozygous TRAP rats and male KSN/nu-nu nude mice were obtained from Oriental Bio Service, Inc. (Kyoto, Japan) and Charles River Japan Inc. (Atsugi, Japan), respectively. All animals were maintained in air-conditioned and specific pathogen-free animal rooms at 23 ± 2 °C and 55 ± 5% humidity under a 12 h light/dark cycle. Experimental diets and water were provided *ab libitum*. All animal experiments were carried out under protocols approved by the Institutional Animal Care and Use Committee of Nagoya City University School of Medical Sciences (no. H30M-25, approved on 1 May 2018 for TRAP rat model, and no. H30M-37, approved on 18 Jun 2018 for PCai1 xenograft model).

### 2.4. Experimental Protocol of TRAP Rat Model

A total of 33 six-week-old TRAP rats were divided into a control group that received an AIN76A powder basal diet (*n* = 12) (Oriental Bio Service, Inc., Kyoto, Japan), and two treatment groups that received 0.2% (*n* = 9) or 1% (*n* = 12) PRE-HIF supplied diet for 10 weeks. After sacrifice, ventral and lateral lobe portions of each rat prostate were frozen and stored at ‒80 °C until processing. Another prostate part was fixed in 10% formalin for paraffin-embedded sections. To assess the development of prostatic neoplastic lesions, the incidence of adenocarcinoma and the relative percentage of neoplastic lesions were quantified using hematoxylin and eosin (H&E)–stained slides as previously described [13,21]. Neoplastic lesions in TRAP rat prostates were histologically classified as low-grade (LG-PIN) or high-grade prostatic intraepithelial neoplasia (HG-PIN), or adenocarcinoma (Figure 1). Testosterone and 17 β-estradiol serum levels of TRAP rats were measured using an enzyme-linked immunosorbent assay kit according to the manufacturer’s instructions (Abcam, Cambridge, UK).

### 2.5. Experimental Protocol of the PCai1 Xenograft Model

PCai1 cells (1 x 10^5^ cells/mouse) were subcutaneously injected into seven-week-old nude mice. One week after transplantation, mice were randomly divided into three groups (*n* = 15 per group): control, 0.2%, and 1% PRE-HIF mixed diets. Tumor sizes and body weights were measured every week. Tumor volume was estimated as follows: 0.52 × length × width × height dimension (millimeters). After six weeks, all mice were sacrificed and primary tumor halves frozen and stored at ‒80 °C for Western blotting. The remaining tumors were fixed in formalin, embedded in paraffin and sectioned.

### 2.6. Immunohistochemistry

Deparaffinized sections were incubated with antibodies against Ki-67 (SP6; Acris Antibodies GmbH, Herford, Germany), AR (Santa Cruz Biotechnology, TX, USA), SV40 Tag (PharMingen, CA, USA), and CD31 (Abcam, Cambridge, UK). To detect apoptotic cells, TUNEL assay using an in situ apoptosis detection kit was performed according to the manufacturer’s instructions (Takara, Otsu, Japan).

### 2.7. Western Blot Analysis

Proteins from frozen tissues or harvested cells were extracted as previously outlined [20]. The antibodies used were against the following antigens: fatty acid synthase (FAS), adenosine monophosphate-activated protein kinase (AMPK)α, phospho-AMPKα, cyclin D1, p21, cleaved caspase 3, caspase 3, cleaved caspase 7, caspase 7, p38 mitogen-activated protein kinase (MAPK), phospho-p38 MAPK, extracellular signal-regulated kinase (ERK)1/2, and phospho-ERK1/2 (Cell Signaling Technology, MA, USA); cdk4 (Thermo Fisher Scientific, MA, USA); prostate-specific antigen (PSA; DAKO, Tokyo, Japan); and AR, NKX3.1, and β-actin (Sigma-Aldrich, St Louis, MO, USA). Each band intensity was semi-quantified using ImageJ 1.51K (National Institute of Health, Bethesda, MD, USA).

### 2.8. Cell Proliferation Assay

The proliferation of prostate cancer cells was determined by colorimetric WST-1 assay. LNCaP or PCai1 cells (3 x 10^3^ cells/well) were seeded in 96-well plates. The cells were then treated with various concentrations of PRE-HIF (50–400 µg/mL), C3G or P3G (5–50 µM), which dissolved in dimethylsulfoxide (DMSO). The DMSO concentration in the conditioned culture medium less than 0.5% were accepted for the experiment. At 24, 48, and 72 h after incubation, 10 µL of WST-1 reagent (Roche, Basal, Switzerland) was added to each well. After that, the absorbance was measured at 430 nm using a SpectraMax^®^ iD3 microplate reader (Molecular Devices, San Jose, CA, USA). The percentage of cell proliferation was calculated relative to the control vehicle (0 µg/mL), and interpreted from three independent experiments.

### 2.9. RNA Extraction, cDNA Preparation, and Quantitative Real-Time PCR

LNCaP or PCai1 cells (2 x 10^5^ cells/well) were seeded and incubated with PRE-HIF. Total RNA from treated cells was isolated using Isogen (Nippon Gene, Tokyo, Japan) and converted to cDNA using PrimeScript™ RT Master Mix (Takara) according to the manufacturer’s instructions. The cDNA template was subjected to quantitative real-time PCR (qRT–PCR) using TB Green™ Premix Ex Taq ™ II (Takara) in an AriaMx Real-Time PCR System (Agilent Technologies, Santa Clara, CA, USA). Primer sequences are listed as follows: human AR, 58°C, 5′-TGTCA ACTCCAGGATGCTCTACTT-3′ and 5′-TTCGGACACACTGGCTGTACA-3′; human cyclin D1, 60°C, 5′-CCGAGAAGCTGTGCAT CTAC-3′ and 5′-CAGGTTCAGGCCTTGCACTG-3′; rat AR, 58°C, 5′-AAGACCTGCCTGATCTGTG GA-3′ and 5′-CTTCAAAAGAGCTGCGGAAG-3′; rat cyclin D1, 60°C, 5′-CAAGTGTGACCCGGA CTGC-3′ and 5′-GCTTCTTCCTCCACTTCCCC-3′; rat GAPDH, 55°C, 5′-GCATCCTGCACCACCA AC-3′ and 5′-GCCTG CTTACCACCT TGTT-3′. Target mRNA levels of treated cells were normalized to GAPDH and expressed relative to control cells.

### 2.10. Cell-Cycle Analysis

LNCaP or PCai1 cells (2 × 10^5^ cells/well) were plated and cultured in conditioned medium with PRE-HIF, bicalutamide, C3G, or P3G. Cells were collected, fixed in 70% ethanol, and sequentially stained with Guava^®^ Cell Cycle reagent (Guava Technologies, Hayward, CA, USA). To analyze cell-cycle phase distributions, stained cells were subjected to Guava^®^ easyCyte flow cytometers using Guava InCyte™ software (Guava Technologies).

### 2.11. Statistical Analysis

Data from the *in vivo* and *in vitro* experiments were presented as mean ± standard deviation (SD). Statistical analysis was performed with IBM SPSS^®^ Statistics version 20.0 (IBM Corp., Armonk, NY, USA) using one-way ANOVA with a post-hoc test: LSD or Dunnett’s test. Significance was considered at *p* < 0.05.

## 3. Results

### 3.1. PRE-HIF Ameliorates Prostate Tumorigenesis in a TRAP Rat Model

Supplementation with PRE-HIF had no significant effect on body, organ weights, daily food intake, serum levels of testosterone and estrogen in TRAP rats (Table 1, Figure 2a,b). During the 10 weeks of the experiment, no one of them showed signs of toxicity. The average HIF intake of rats in 0.2% HIF and 1% HIF mixed diet group was 97.7 ± 7.1 and 505.0 ± 29.6 mg/kg/day, respectively (Table 1). In ventral prostates, the percentage of adenocarcinoma in the PRE-HIF–fed group was lower than that of the control group, but did not reach statistical significance (Table 2; (F (2, 30) = 2.330, *p* = 0.075). However, a dose-dependent effect of PRE-HIF on prostatic lesions was observed in rat lateral prostates (Table 2 and Figure 2c), which is functionally identical to the peripheral zone of the human prostate [22]. The percentage of adenocarcinoma in the lateral prostates was significantly decreased in the 1% PRE-HIF-fed group (F (2, 30) = 2.748, *p* = 0.049). Conversely, LG-PIN was significantly increased by 1% PRE-HIF consumption, indicating most acini in the lateral prostates persisted in a benign neoplastic stage (F (2, 30) =4.779, *p* = 0.009). These alteration by PRE-HIF resulted in significant suppressed incidence of adenocarcinoma (F (2, 30) = 4.636, *p* = 0.012). These results suggest that PRE-HIF suppresses prostate carcinogenesis in TRAP rats without altering serum testosterone and estrogen levels.

### 3.2. PRE-HIF Down-Regulates the Expression of Androgen Receptor and Cellular Growth and Metabolism Proteins in Lateral Prostates of TRAP Rats

Significantly decreased AR expression was observed immunohistologically in lateral prostates of the PRE-HIF supplied group (Figure 2c,d). The expression of AR and NKX3.1 (a collaborative factor of AR) [23] in lateral prostates was greatly down-regulated in PRE-HIF-fed rats, while no significant difference in SV40T expression was exhibited as shown by western blotting (Figure 3a). In addition, the Ki-67 labeling index was significantly decreased in lateral prostates of PRE-HIF-fed compared to control rats (Figure 3b; *p* < 0.01). However, the apoptotic index analyzed by TUNEL staining was not significantly different between control and treatment groups (Figure 3c). Proteins in cell growth pathways, such as cyclin D1, cdk4, and p-p38 MAPK was down-regulated in lateral prostates of the PRE-HIF-fed group (Figure 3a). Although the down-regulation of p-ERK1/2 was observed in the 1% PRE-HIF supplied group, it failed to reach statistical significance. Consistent with TUNEL assay results, PRE-HIF had no effect on apoptotic proteins in lateral prostates (Appendix A). Moreover, PRE-HIF also strongly down-regulated proteins related to cellular metabolism, including FAS and pAMPKα (Figure 3a). Therefore, these results suggest the inhibitory effect of PRE-HIF in the TRAP model involves pathways responsible for cellular growth and metabolism, but not cell apoptosis.

### 3.3. PRE-HIF Retards Tumor Growth and Inhibits Cell Growth Pathways in PCai1 Xenograft Model

Next, we investigated whether PRE-HIF affected the tumor growth of CRPC using rat CRPC xenograft model of PCai1. Mice in 0.2% and 1% PRE-HIF group were received 248.8 ± 24.5 and 1233.6 ± 46.3 mg/kg/day of PRE-HIF, respectively. In comparison with the control diet, a 1% PRE-HIF supplied diet significantly decreased tumor size (Figure 4a; *p* < 0.01) and suppressed tumor growth in PCai1 xenograft mice (Figure 4b). A significant decrease in AR expression (*p* < 0.001) and the Ki-67 labeling index (*p* < 0.01) was observed in PCai1 tumors of the PRE-HIF-fed group (Figure 4c). However, no significant differences in the apoptotic index and vessel number between control and PRE-HIF supplied groups were found (Figure 4c). Proteins involved in cell growth pathways were down-regulated in tumors of PRE-HIF-fed groups (Figure 5), while cleaved caspase 3 and 7 protein levels remained unchanged (Appendix A). Similar to the TRAP model, PRE-HIF down-regulated FAS expression and attenuated AMPKα activation in PCai1 xenografts (Figure 5). These findings suggest a potential inhibitory action of PRE-HIF on pathways related to cell growth and energy deprivation.

### 3.4. PRE-HIF Suppresses Cell Proliferation and Induces Cell-Cycle Arrest in Prostate Cancer Cell Lines

In the androgen-dependent prostate cancer cell line, LNCaP, PRE-HIF significantly suppressed cell proliferation (Figure 6a; *p* < 0.01), increased the proportion of cells in the G0/G1 phase, and decreased the proportion of cells in S and G2/M phases (Figure 6b; *p* < 0.05 for all). In addition, cell proliferation by the androgen-independent prostate cancer cell line, PCai1, tended to decrease with PRE-HIF treatment (Figure 6c; *p* < 0.01). Cell-cycle analysis of PCai1 showed significantly increased proportion of G0/G1-phase cells (*p* < 0.01) and a decreased proportion of G2/M-phase cells (*p* < 0.05) when cells were incubated with PRE-HIF (Figure 6d). Interestingly, the more potent effects of PRE-HIF regarding the inhibition of cell proliferation and induction of G1 cell-cycle arrest were observed under hormone-depleted conditions (Appendix A).

To emphasize the modulating effects of PRE-HIF on AR signaling, the expression of AR, prostate-specific antigen (PSA), and NKX3.1 were determined in LNCaP cells. Decreased AR protein levels and pivotal downstream targets were found in PRE-HIF-treated LNCaP cells (Figure 6e). Additionally, proteins responsible for cell growth were down-regulated by PRE-HIF treatment (Figure 6e,f), but apoptotic protein expression remained unaffected (Appendix A). In both cell lines, PRE-HIF also decreased AR and cyclin D1 mRNA levels in a dose-dependent manner (Figure 6g–j), indicating PRE-HIF regulated AR and cyclin D1 at the transcriptional level. Furthermore, PRE-HIF tended to down-regulate FAS and pAMPKα proteins in both LNCaP and PCai1 cells (Figure 6e,f). These findings confirm the modulating effects of PRE-HIF on AR signaling, cell growth pathways, and cell metabolism.

### 3.5. Cyanidin-3-Glucoside, but Not Peonidin-3-Glucoside, Modulates AR Expression and Attenuates ERK1/2 and p38 MAPK Activation in PCai1 Cells

To identify whether the main anthocyanins in PRE-HIF, C3G, and P3G [8] are active compounds, the effects of both anthocyanins were examined in PCai1 cells. In the same line previously treated with PRE-HIF, C3G decreased the proliferation of PCai1 cells in a dose-dependent manner, but P3G had no effect (Figure 7a,b). Consistently, C3G significantly increased G0/G1-phase cells and decreased G2/M-phase cells, while the proportions of PCai1 cells in various cell-cycle phases were unaffected by P3G treatment (Figure 7c,d). Proteins involved in cell growth, including AR, cyclin D1, p-ERK1/2, and p-p38 MAPK were down-regulated in C3G-treated cells (Figure 7e). In contrast, the expression of these proteins remained the same with P3G treatment (Figure 7f). However, both C3G and P3G down-regulated FAS expression (Figure 7e,f), suggesting a synergistic effect on energy deprivation. With regard to the structural difference between C3G and P3G, it has been suggested that hydroxyl groups may be involved with biological effects [21]. Collectively, we suggest that C3G is the main active compound of PRE-HIF in the inhibition of prostate cancer cell growth.

## 4. Discussion

Recently, the chemopreventive effects of natural or synthetic substances have described extensively. Anthocyanins, the subunit substance of flavonoids, have become interesting phytochemicals which exert anti-cancer effects against various cancers. Tanaka et al. reported that anthocyanins suppressed angiogenesis by inhibiting VEGF-induced proliferation and migration of HUVECs [24]. A purified anthocyanin, including C3G from blackberries and mulberry anthocyanins, was reported to inhibit lung carcinogenesis both *in vitro* and *in vivo* [25,26]. Similarly, it has been revealed that major anthocyanins, C3G and pelargonidin-3-glucoside (Pg3G) in purple corn color retarded the proliferation of human prostate cancer cell line in a dose-dependent manner [21]. Here, we interested to explore the effects of PRE-HIF, an anthocyanin-rich extract fraction from purple rice extract [8,9], on prostate carcinogenesis and CRPC. As previously reported, hexane effectively removed tocols and gamma-oryzanol from the parent ethanolic extract of purple rice (PRE) [9]. For this reason, anthocyanins including C3G and P3G are primarily concentrated in the hexane insoluble part PRE-HIF [8,9].

Firstly, we investigated the protective effects of PRE-HIF against prostate carcinogenesis by using the TRAP rat model. PRE-HIF suppressed progression of prostate carcinogenesis without any toxicity *in vivo*. The markedly down-regulation of AR and NKX3.1 was shown in the lateral prostates of TRAP rats. The effect was also observed in rat prostate BPH treated PRE-HIF [9], strongly indicating the potential effects of PRE-HIF on AR signaling modulation. AR pathway is still important signaling after progression of CRPC. CRPC cells can maintain a dependence on AR signaling and even acquiring resistance to hormonal interventions [27]. Therefore, we further investigated the effects of PRE-HIF on CRPC by using rat-derived CRPC cell line PCai1. Similar to TRAP model, PRE-HIF retarded the tumor growth in PCai1 xenograft model. A lower AR expression was observed in PCai1 tumors from high-dose PRE-HIF consuming group. These results indicate that PRE-HIF induced AR suppresses not only early prostate carcinogenesis but also the late stage with castration-resistance.

In the present study, PRE-HIF down-regulated the expression of proteins in cell growth pathways, while no changes in caspases level in prostate cancer. PRE-HIF reduced Ki-67 index by more than 20% (from 79.7 ± 16.8% to 54.3 ± 18.8%) in lateral prostate of TRAP model, while it did not alter the apoptotic index (control: 3.8 ± 1.0%, 1% PRE-HIF: 3.3 ± 1.4%, respectively). In accordance with previous studies, TUNEL indices were increased by around 5%–6%, at least less than 10% with induction of apoptosis by test chemicals in prostate cancer tissue [13,17]. This indicates that a strong anti-proliferative effect may provide chemopreventive or chemotherapeutic effects without induction of apoptosis.

In addition to anti-proliferative and anti-apoptotic effects, anti-angiogenesis is one of the underlying mechanisms that can suppress the tumor growth and progression [28]. A range of natural or synthetic compounds inhibits cancer metastasis by blocking the formation of blood vessels [24,29,30]. In this study, the effect of PRE-HIF on the vessel numbers was not observed in PCai1 tumor. This suggests that anti-tumor effect of PER-HIF is mainly due to its anti-proliferative activity. Considering that main constituents of PRE-HIF are anthocyanins, it may have potentials to suppress cancer invasion and metastasis by targeting to other mechanisms, such as inhibition of cancer cell migration [31]. Hence, the study of anti-invasive and anti-metastatic effects of purple rice extract is still challenging and interesting, according to the obvious inhibitory effects on the progression of prostate carcinogenesis and CRPC.

Since the TRAP model is representative of an androgen-dependent prostate cancer [10,11], we selected an androgen-dependent human prostate cancer cell line, LNCaP, to explore the molecular mechanism(s) of PRE-HIF. The down-regulated expression of AR, PSA, and NKX3.1 in PRE-HIF treated LNCaP cells highlighted the modulating effect of PRE-HIF on AR signaling. There findings suggest that PRE-HIF has a suppressive effect in early stage of prostate cancer both *in vitro* and *in vivo*. Similarly, PRE-HIF down-regulated AR expression in the rat CRPC cell line, PCai1. Although PCai1 cell growth was androgen-independent, increased AR expression was clearly observed compared to human prostate cancer cell lines [18]. In addition, AR amplification is associated with resistance to ADT and promotes CRPC progression [32]. Altogether, in addition to a benign stage, AR remains an important therapeutic target for advanced prostate cancer.

Apart from AR signaling, metabolic reprogramming in prostate cancer cells can promote cancer development [33]. AMPK, a master regulator of cellular metabolism, is a survival factor for prostate cancer [34]. Although previous studies initially reported on tumor suppressor function, an oncogenic role for AMPK in prostate cancer has recently been revealed. The expression of pAMPK was elevated in human prostate cancer cells and correlated with disease stages in patient clinical samples [35,36,37]. Additionally, AMPKα knock-down suppresses proliferation of prostate cancer cell lines [38]. In addition to glucose, proliferating cancer cells increase the need for lipid fuels [39]. The overexpression of lipid-metabolizing enzymes at all stages of prostate cancer indicated their active roles in disease progression [40]. FAS, a key lipid-producing enzyme, tends to be up-regulated from low grade PIN to invasive carcinoma [41]. In AR-expressing human prostate adenocarcinoma cells, the forced expression of FAS promotes soft agar growth and tumor formation [42]. Thus, FAS and AMPKα may be potential therapeutic targets in prostate cancer. In this study, we showed that PRE-HIF down-regulated FAS and pAMPKα expression in lateral prostates of TRAP rats, PCai1 tumors of a xenograft model, and prostate cancer cell lines. Thus, beyond its role in AR signaling and cell growth pathways, PRE-HIF also altered metabolic pathways *in vitro* and *in vivo*.

Taken together, we conclude PRE-HIF can retard prostate cancer and CRPC growth. Its potential inhibitory effect involves pathways that account for cell growth and energy deprivation by modulating AR and MAPK expression. Therefore, PRE-HIF may be a promising chemopreventive and therapeutic agent for prostate cancer.

## Figures and Tables

**Figure 1 nutrients-12-00558-f001:**
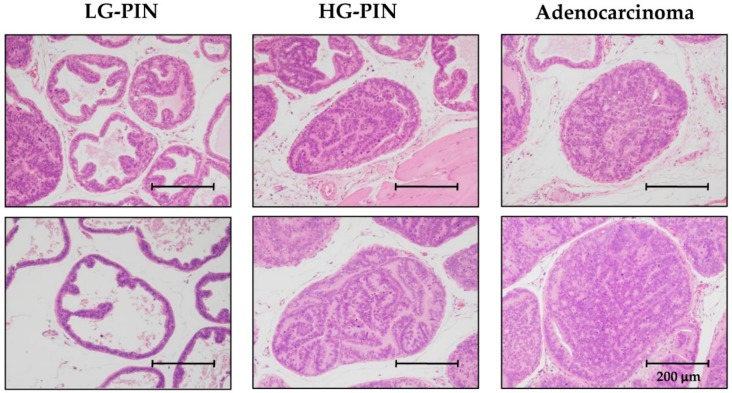
Prostatic proliferative lesions that were divided into 3 types: low grade (LG-PIN), high-grade prostatic intraepithelial neoplasia (HG-PIN), and adenocarcinoma (Magnification, 200×; scale bar, 200 µm).

**Figure 2 nutrients-12-00558-f002:**
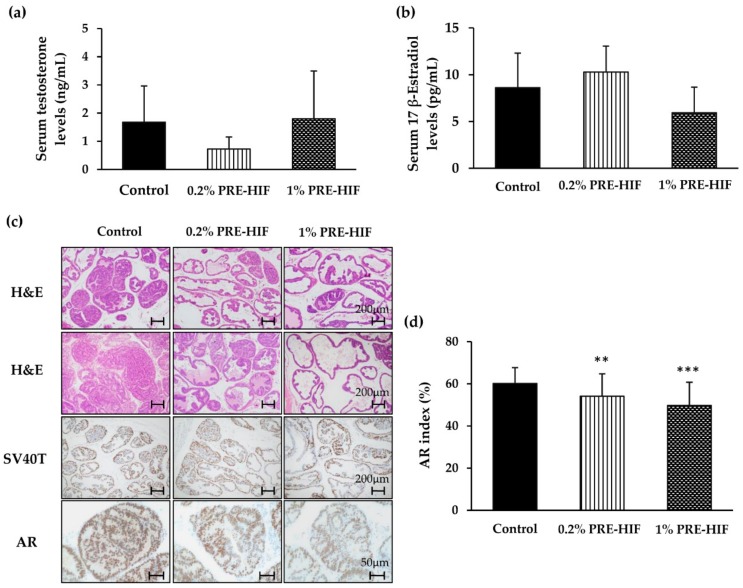
Effects of PRE-HIF on prostate carcinogenesis in TRAP rats. (**a**) Levels of serum testosterone and (**b**) 17 β-estradiol. (**c**) Hematoxylin and eosin (H&E) staining, SV40T antigen expression by immunostaining in the lateral prostate (magnification, 100×; scale bar, 200 µm) and immunostained androgen receptor (AR) images of high-grade prostatic intraepithelial neoplasia in lateral prostates (magnification, 400×; scale bar, 50 µm). (**d**) Immuno-labeling indices for AR-positive cells relative to viable tumor cells. Data are shown as mean ± standard deviation (SD). ** *p* < 0.01; *** *p* < 0.001 compared to control group. PRE-HIF, hexane insoluble fraction of purple rice ethanolic extract.

**Figure 3 nutrients-12-00558-f003:**
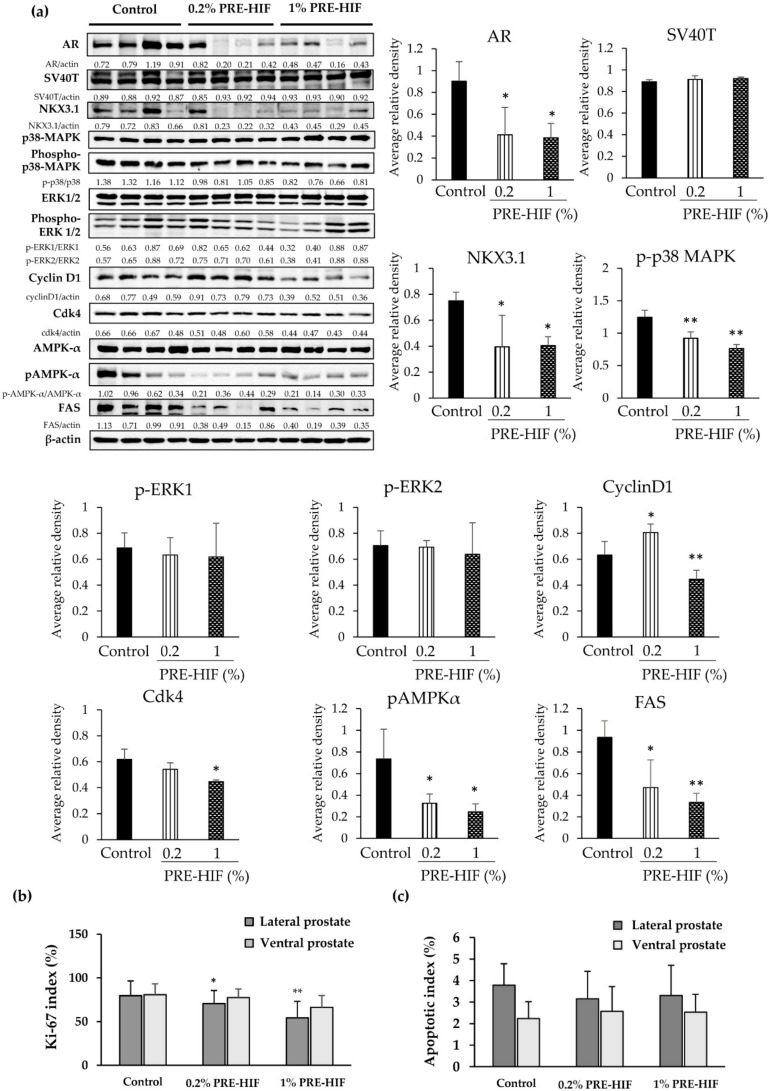
Effects of PRE-HIF on protein expression, Ki-67 and apoptotic index of rats’ prostates. (**a**) Western blots and average relative band density of proteins related to cellular growth and metabolism using four random lateral prostate samples from Transgenic Rat for Adenocarcinoma of Prostate (TRAP) rats. (**b**) Labeling indices for Ki-67- and (**c**) terminal deoxynucleotidyl transferase dUTP nick end labeling (TUNEL)-positive cells in the prostates (samples from all rats were used for analysis). Data are shown as mean ± standard deviation (SD). * *p* < 0.05; ** *p* < 0.01 compared to the control group. PRE-HIF, hexane insoluble fraction of purple rice ethanolic extract.

**Figure 4 nutrients-12-00558-f004:**
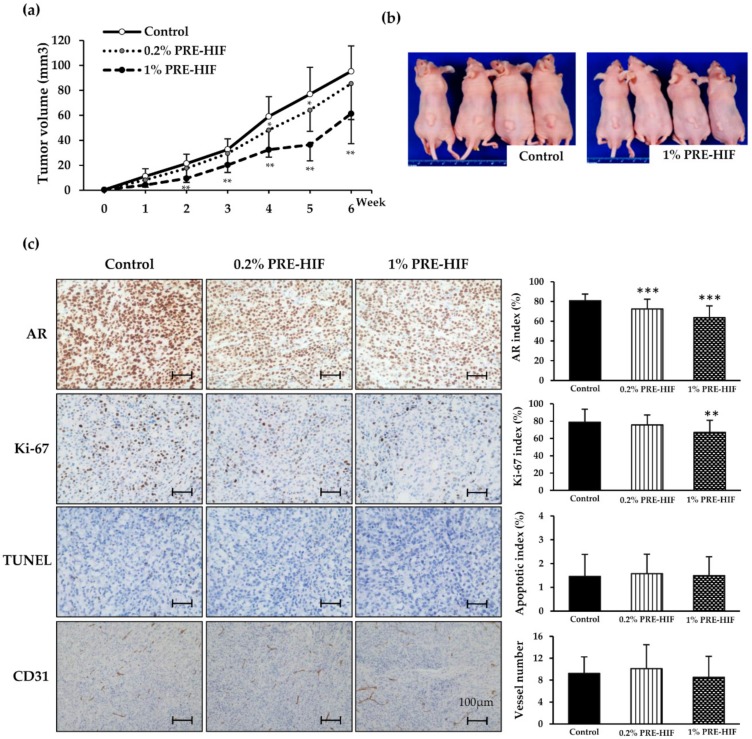
Effects of PRE-HIF on CRPC in PCai1 xenograft model. (**a**) Growth curve of tumors. (**b**) Photographs of PCai1 xenograft mice in control diet and 1% PRE-HIF mixed diet groups at the end point of the experiment. (**c**) PCai1 tumor sections were immunohistochemically staining for androgen receptor (AR), Ki-67, terminal deoxynucleotidyl transferase dUTP nick end labeling (TUNEL) assay, and the blood vessel marker, CD31 (scale bar, 100 µm). Samples of all mice were used for analysis (*n* = 15 per group). Data in all bar charts are shown as mean ± standard deviation (SD). * *p* < 0.05; ** *p* < 0.01; *** *p* < 0.001 compared to control group. PRE-HIF, hexane insoluble fraction of purple rice ethanolic extract; CRPC, castration-resistant prostate cancer.

**Figure 5 nutrients-12-00558-f005:**
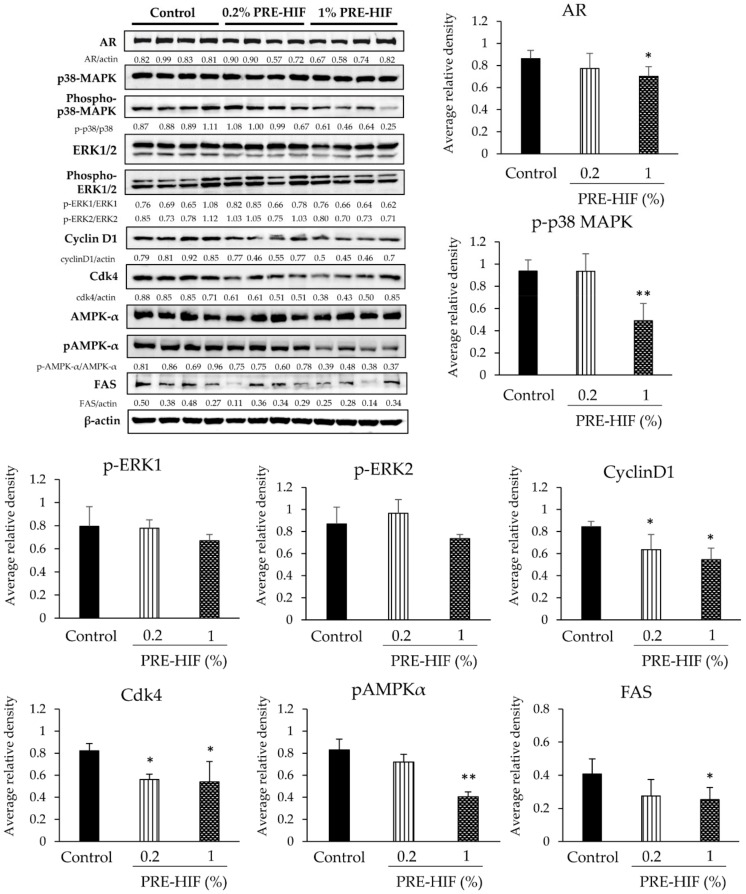
Effects of PRE-HIF on protein expression in PCai1 xenograft model. Western blot and average relative band density of proteins related to cellular growth and metabolism using four random tumor samples from the PCai1 xenograft model. Data in all bar charts are shown as mean ± standard deviation (SD). * *p* < 0.05; ** *p* < 0.01 compared to control group. PRE-HIF, hexane insoluble fraction of purple rice ethanolic extract.

**Figure 6 nutrients-12-00558-f006:**
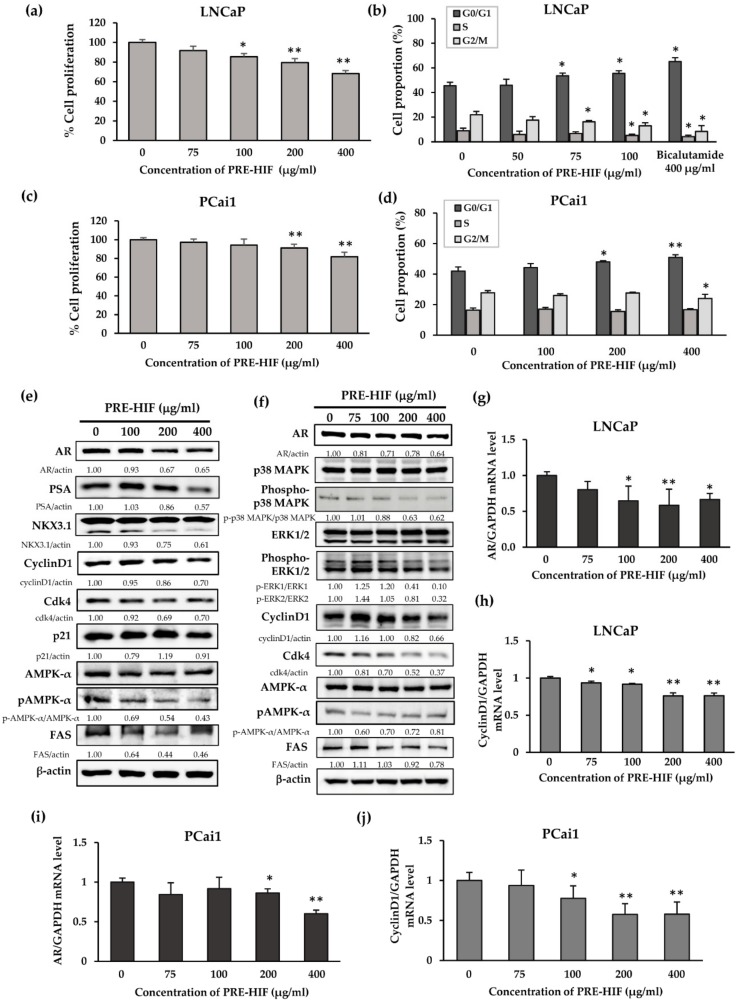
Effects of PRE-HIF on prostate cancer cell lines. (a) Cell proliferation of LNCaP and (c) PCai1 cells treated with various concentrations of PRE-HIF for 72 h. (b) Effects on the cell cycle of LNCaP and (d) PCai1 cells treated with various concentrations of PRE-HIF for 72 h. (e) Immunoblots of protein lysates from PRE-HIF treated LNCaP and (f) PCai1 cells (72 h) were probed with antibodies against proteins related to cell growth and metabolism. At 48 h of PRE-HIF incubation, androgen receptor (AR), and cyclin D1 mRNA levels in LNCaP (g, h) and PCai1 (i, j) cells were analyzed by quantitative real-time (RT)–PCR. All data in bar charts are shown as the mean ± standard deviation (SD) of three independent experiments. * *p* < 0.05; ** *p* < 0.01 compared to control. PRE-HIF, hexane insoluble fraction of purple rice ethanolic extract.

**Figure 7 nutrients-12-00558-f007:**
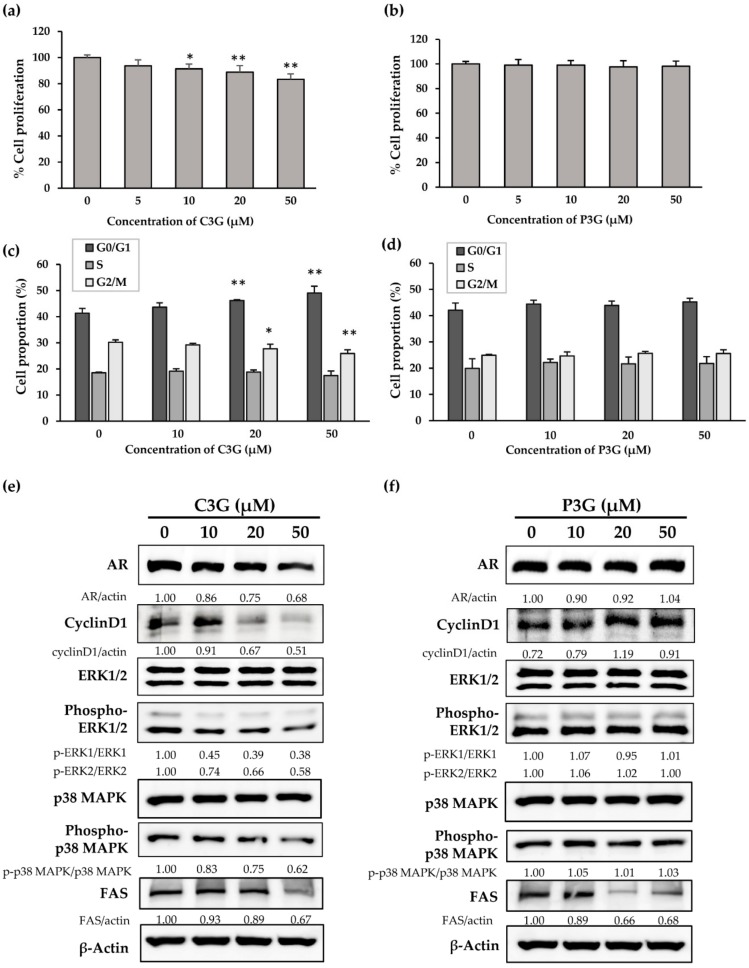
Effect of anthocyanins on PCai1 rat CRPC cell line. (**a**) Cell proliferation of PCai1 cells after incubation with C3G or (**b**) P3G. Effects of (**c**) C3G or (**d**) P3G on the cell cycle of PCai1 cells (72 h). (**e**) Immunoblots of protein lysates from PCai1 cells treated with C3G or (**f**) P3G for 72 h. All data in bar charts are shown as the mean ± standard deviation (SD) of three independent experiments. * *p* < 0.05; ** *p* < 0.01 compared to control. C3G, cyanidin-3-glucoside; P3G, peonidin-3-glucoside; CRPC, castration-resistant prostate cancer.

**Table 1 nutrients-12-00558-t001:** Body weights, organ weights, and PRE-HIF intake of TRAP rats treated with PRE-HIF.

Trait	Group		
	Control	0.2% PRE-HIF	1% PRE-HIF
**No. of rats**	12	9	12
**Initial body weights** (**1st Day**) (**g**)	180.5 ± 22.5	180.2 ± 31.8	180.3 ± 26.9
**Final body weights** (**70th Day**) (**g**)	609.5 ± 62.7	600.1 ± 65.9	617.2 ± 72.8
**Average food intake** (**g/rat/day**)	30.0 ± 1.8	29.3 ± 2.1	31.2 ± 1.8
**Average PRE-HIF intake** (**mg/kg/day**)	0	97.7 ± 7.1	505.0 ± 29.6
**Organ weights** (**g**)			
**Liver**	20.7 ± 4.9	19.1 ± 3.1	20.9 ± 3.1
**Kidney**	3.0 ± 0.4	3.1 ± 0.3	3.1 ± 0.3
**Testis**	3.8 ± 0.3	3.7 ± 0.3	3.7 ± 0.3
**Ventral prostate**	0.27 ± 0.06	0.31 ± 0.07	0.27 ± 0.07

Data are shown as the mean ± standard deviation (SD). TRAP, Transgenic Rat for Adenocarcinoma of Prostate; PRE-HIF, hexane insoluble fraction of purple rice ethanolic extract

**Table 2 nutrients-12-00558-t002:** Incidence of carcinoma and quantitative evaluation of prostatic neoplastic lesions in TRAP rats treated with PRE-HIF.

Treatment	No. of Rats	Incidence of Adenocarcinoma (%)	% of Lesion in Prostates ^a^		
			LG-PIN	HG-PIN	Adenocarcinoma
**Ventral prostates**					
**Control**	12	12 (100%)	7.2 ± 3.1	81.4 ± 7.2	11.4 ± 6.5
**0.2% PRE-HIF**	9	9 (100%)	10.5 ± 4.1	81.2 ± 4.1	8.3 ± 5.0
**1% PRE-HIF**	12	11 (92%)	8.3 ± 4.6	85.1 ± 4.6	6.6 ± 4.2
**Lateral prostates**					
**Control**	12	12 (100%)	8.4 ± 2.9	85.4 ± 5.2	6.5 ± 3.7
**0.2% PRE-HIF**	9	6 (67%)	11.3 ± 5.7	84.7 ± 5.3	4.0 ± 3.5
**1% PRE-HIF**	12	6 (50%) *	16.3 ± 8.1 **	81.0 ± 7.4	2.7 ± 3.7 *

TRAP, Transgenic Rat for Adenocarcinoma of Prostate; PRE-HIF, hexane insoluble fraction of purple rice ethanolic extract; LG, low grade; HG-PIN, high grade prostatic intraepithelial neoplasia. ^a^; Values expressed as the mean ± standard deviation (SD). * *p* < 0.05; ** *p* < 0.01 compared to control group (one-way ANOVA and Dunnett’s post-hoc test).

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
