# Peer review of "Hexane Insoluble Fraction from Purple Rice Extract Retards Carcinogenesis and Castration-Resistant Cancer Growth of Prostate Through Suppression of Androgen Receptor Mediated Cell Proliferation and Metabolism"

_nutrients, 2020, doi:10.3390/nu12020558_

Round 1

Reviewer 1 Report

In this paper, Ranchana and colleagues seek to demonstrate the protective effects of hexane insoluble fraction from purple rice extract (PRE-HIF) against prostate cancer. They claim PRE-HIF can inhibit the proliferation and metabolic pathway of prostate cancer cells. The novelty of this paper is a major issue since a similar finding has already been reported (Kiriya, Chanarat, et al. "Purple rice extract inhibits testosterone‐induced rat prostatic hyperplasia and growth of human prostate cancer cell line by reduction of androgen receptor activation." Journal of food biochemistry 43.9 (2019): e12987.). The authors also cited this paper to support their hypothesis while they seemed to ignore that the growth inhibition effect of purple rice extract to prostate cancer cell line has been demonstrated in this paper.

Below please find my specific comments.

Since the authors did not provide literature to support their extraction methods, they should provide data on the extracted products, such as yield and contents. At the same time, the percentage diet is not precise enough to accurately reflect the actual amount accepted by experimental animals. Some results are not convincing. For example, the one-way ANOVA test showed no significant difference in Table 2 based on the data of mean, SE and case numbers. The authors should provide the statistics like F value rather than just ranges of p values. As shown in some figure of immunoblots, the protein level changes of some genes are either very minor or lack consistency. As mentioned in the text, the TRAP rats were divided randomly into three groups. So, why the group of 0.2% PRE-HIF only included 9 animals? The authors should also provide the numbers of nude mice included in PCai1 xenograft model. In Figure 1, the authors showed the different types of prostatic proliferative lesions. However, I think readers are more interested in images from different experimental groups.

5. In line 162, Table 1 should be Table 2. In section 3.5, I believe all the results should be presented in Figure 5.

Author Response

Comment from reviewer 1: Thank you very much for giving us the opportunity to revise our manuscript. We appreciate your kind comment and suggestion. We hope that the quality of our work would be improved after revision.

A list of comments

1) Since the authors did not provide literature to support their extraction methods, they should provide data on the extracted products, such as yield and contents.

Response: 

              Thank you very much for pointing this out. We do apologize for our obscure statement. The extraction method, yield and anthocyanin content of PRE-HIF has published. The yields of PRE-HIF were 2.16 g per 100 g purple rice grain. Anthocyanins including cyanidin-3-glucoside (C3G) and peonidin-3-glucoside (P3G) in PRE-HIF was 4.87 ± 0.05 and 2.25 ± 0.02 mg/g extract, respectively [a new reference #9, as indicated below]. In accordance with your suggestion, we added the reference and descriptions of extraction method in chapter 2.1.

Reference

Yeewa, R.; Sakuludomkan, W.; Kiriya, C.; Khanaree, C.; Chewonarin, T. Attenuation of benign prostatic hyperplasia by hydrophilic active compounds from pigmented rice in testosterone implanted rat model. Food & Function 2020, https://doi.org/10.1039/C9FO02820J.

2) At the same time, the percentage diet is not precise enough to accurately reflect the actual amount accepted by experimental animals.

Response: 

              We agree with you that the percentage of PRE-HIF in the diet did not represent the actual PRE-HIF intake of the rats. Thus, the average PRE-HIF intake was calculated by using the data of rats’ or mice’ food intake and percentage of PRE-HIF in the diet. We have added data for rats in chapter 3.1 and modified Table 1, and data for mice in chapter 3.3.

3) Some results are not convincing. For example, the one-way ANOVA test showed no significant difference in Table 2 based on the data of mean, SE and case numbers. The authors should provide the statistics like F value rather than just ranges of p values.

Response: 

              As suggested by reviewer, we have added the F value of ANOVA, in addition to p value in histological analysis of TRAP as shown in chapter 3.1.

4) As shown in some figure of immunoblots, the protein level changes of some genes are either very minor or lack consistency.

Response: 

              As mentioned by reviewer 1, results from in vivo assay sometimes fail to exhibit consistency, due to the individual differences among animals. To highlight and address the down-regulation of proteins, we measured the intensity of each band of western blotting and then relative density score in each group was statistically analyzed. The results show that AR, NKX3.1, p-p38 MAPK, cyclinD1, cdk 4, p-AMPKα and FAS in the lateral prostates of TRAP rats and PCai1 xenografts were significantly down-regulated by 1% PRE-HIF. On the other hand, the down-regulation of pERK1/2 in both prostate tissue and PCai1 tumor was not significant. We have added these quantitative data in new Figure 3 (TRAP) and new Figure 5 (PCai1 xenograft) and modified description about the expression of ERK1/2 in Abstract and chapter 3.2.

5) As mentioned in the text, the TRAP rats were divided randomly into three groups. So, why the group of 0.2% PRE-HIF only included 9 animals?

Response: 

              As mentioned by reviewer 1, sample number is a crucial component of any research, including animal studies. In the present study, the control and high dose group contain 12 TRAP rats per group, but 9 rats subjected to a low dose group. The reason why fewer animals in low dose group was the limitation of total TPAP rat numbers we could prepare. We consider that 12 animals provide significant effects of a test chemical to avoid influences of individual differences among animals. Therefore, from total 33 rats, we decided to use 12 animals for the control or high dose group, and residual 9 rats, the number is more than the minimum numbers required in animal research, were divided for low dose group. With reflect these, descriptions of experimental protocol of TRAP rat in chapter 2.4 have been modified.

6) The authors should also provide the numbers of nude mice included in PCai1 xenograft model.

Response: 

              For PCai1 xenograft model, the numbers of nude mice per group is 15 mice. As suggested by reviewer 1, we have added the numbers of nude mice per group as shown in chapter 2.5.

7) In Figure 1, the authors showed the different types of prostatic proliferative lesions. However, I think readers are more interested in images from different experimental groups.

Response: 

              In Figure 1, we show representative images of each prostatic proliferative lesions in the lateral prostate lobe of TRAP from the control group and clearly indicate the histological criteria that we used for the quantitative evaluation (Table 2). In order to provide better understanding of the histological classification and differences among groups, we have added the additional images of prostatic lesions (Figure 1) and representative analysis results of each group (Figure 2c).

8) In line 162, Table 1 should be Table 2. In section 3.5, I believe all the results should be presented in Figure 5.

Response:

               As suggested by reviewer 1, we have corrected the table or figure number the line 162 and section 3.5.

Reviewer 2 Report

In this paper the authors tested the anticancer effect of a hexane insoluble fraction from a purple rice ethanol extract on different in vivo and in vitro models: a Transgenic Rat for Adenocarcinoma of Prostate (TRAP) model, a rat CRPC xenograft of PCai1 cells, and a cells culture of LNCaP or PCai1. The highlighted antitumor effects are attributed by the authors essentially to the action of Cyanidin-3-O-glucoside, while peonidin-3-O-glucoside does not seem to produce the same effects. The conclusions are that the potential inhibitory effect of hexane insoluble fraction involves pathways that account for cell growth and energy deprivation, in addition to a role in AR signalling.

The paper is well written and the experiments well conducted. The topic is interesting I recommend the publication after minor revision.

The title is not consistent with the conclusions in the abstract and in the main text. The Authors state that PRE-HIF beyond its role in AR signalling and cell growth pathways, blocks the development of prostate cancer and CRPC through inhibition of cell proliferation and metabolic pathways. I would mention this also in the title. The Authors should titrate the PRE-HIF in terms of content in Cyanidin-3-O-glucoside chloride (C3G) and peonidin-3-O-glucoside. Even better would be a metabolomics characterization of the PRE-HIF at least in terms of other antioxidants metabolites. Due to the conclusions of this paper, it could be interesting to have also the evaluation of Cyanidin-3-O-glucoside levels in plasma or tumour at the end of the experiment.

In breef:

The authors should better explain the meaning of 1% PRE-HIF supplied diet. How did they prepare this? How much diet consumed the rats during the 10weeks treatment?? Is possible to have a mean estimation? Chapter 2.1 Please add the exact volumes of the solvents used and the exact weight of the rice extracted. Describe better the procedure. Chapter 2.8 Explain better the experiment. In which solvent was dissolved the PRE-HIF extract and how was calculated the ug/mL concentration? From how much rice was obtained the PRE-HIF extract? Chapter 3.5 Pag. 8 row 255. Fig.4 maybe intended as fig. 5? page 8 Row 259 Avoid potently, some proteins do not seem so downregulated page 3 row 130 Please add the quantity (volumes and concentrations) of PRE-HIF. Add also the range of concentration used for C3G, or P3G added. Figure 2 E. Phospho ERK ½ does not seem so down regulated Figure 3D. The down regulation of proteins P38MAPK, ERK1/2, phosphor ERC1/2, AMPKalfa, FAS, do not seem so significantly. Figure 4E and 4F. The downregulation of most proteins is not so strikingly: see p21, AMPKalpha, FAS.

Author Response

Comment from reviewer 2: Thank you for your thoughtful in reviewing of our revised manuscript. We appreciate your careful checking and kind comments. These would be beneficial for the improvement of our work. The response and modifications are as follows according to your comments.

A list of comments

1) The title is not consistent with the conclusions in the abstract and in the main text. The Authors state that PRE-HIF beyond its role in AR signalling and cell growth pathways, blocks the development of prostate cancer and CRPC through inhibition of cell proliferation and metabolic pathways. I would mention this also in the title.

Response: 

              We agree with you that the title is not completely represent our results. Thus, we have revised the title to the new title “Hexane insoluble fraction from purple rice extract retards carcinogenesis and castration-resistant cancer growth of prostate through suppression of androgen receptor mediated cell proliferation and metabolism”.

2) The Authors should titrate the PRE-HIF in terms of content in Cyanidin-3-O-glucoside chloride (C3G) and peonidin-3-O-glucoside. Even better would be a metabolomics characterization of the PRE-HIF at least in terms of other antioxidants metabolites.

Response: 

              There are some candidates of bioactive compounds of purple rice. In our recent study [a new reference #9, as indicated below], crude ethanolic purple rice extract (PRE) was partitioned with hexane by using a phase to phase extraction. The two phases then being separated into a hexane soluble fraction (PRE-HSF) and a hexane insoluble fraction (PRE-HIF). PRE-HIF contained anthocyanins, P3G and C3G. On the other hand, PRE-HSF contained other antioxidant metabolites including vitamin D derivatives and gamma-oryzanol but no anthocyanins. Furthermore, the anthocyanin-rich fraction, PRE-HIF, was identified as the high active and non-toxic fraction of purple rice extract on the benign prostatic hyperplasia (BPH) setting. Therefore, we are interested in the purple rice anthocyanins as a candidate of prostatic chemopreventive agent. As suggested by reviewer 2, we have added a reference and descriptions in chapter 1 (Introduction).

Reference

Yeewa, R.; Sakuludomkan, W.; Kiriya, C.; Khanaree, C.; Chewonarin, T. Attenuation of benign prostatic hyperplasia by hydrophilic active compounds from pigmented rice in testosterone implanted rat model. Food & Function 2020, https://doi.org/10.1039/C9FO02820J.

3) Due to the conclusions of this paper, it could be interesting to have also the evaluation of Cyanidin-3-O-glucoside levels in plasma or tumour at the end of the experiment.

Response: 

              We agree with an idea of reviewer 2, these would be interesting to detect C3G in serum or tumor. Unfortunately, we could not evaluate levels of C3G in the plasma or tumor, due to the limitation of our experimental design. However, your suggestions are beneficial for our further studies to improve even more the quality and novelty of the researches.

4) The authors should better explain the meaning of 1% PRE-HIF supplied diet. How did they prepare this? How much diet consumed the rats during the 10weeks treatment?? Is possible to have a mean estimation?

Response: 

              The preparation of PRE-HIF mixed powder diet was processed by Oriental MF, Oriental Yeast, Tokyo, Japan. “1% PRE-HIF supplied diet” refers to the concentration of PRE-HIF in the diet, not PRE-HIF intake of animals. According to your advice, we have added the average food and PRE-HIF intake in Table 1, and have modified descriptions in chapter 3.1 and 3.3. The intake of PRE-HIF in TRAP rats was similar to that in the BPH model previously (0.1g - 1g/kg/day, reference #9).

5) Chapter 2.1 Please add the exact volumes of the solvents used and the exact weight of the rice extracted. Describe better the procedure.

Response: 

              In the present study, one kilogram of purple rice grain was roughly blended and then extracted in 1:5 ratios with 80% ethanol by stirring overnight. The ethanol extract was filtered through Whatman filter paper No.1 followed by evaporated using a rotating evaporator (40ºC). After that, the concentrated fraction was partially purified with an equal volume of hexane generating a hexane soluble fraction (PRE-HSF) and hexane insoluble fraction (PRE-HIF). Then, both fractions were evaporated (50ºC), lyophilized and subsequently kept in -20ºC until used. As suggested by reviewer 2, we have modified description of the extraction method in chapter 2.1.

6) Chapter 2.8 Explain better the experiment. In which solvent was dissolved the PRE-HIF extract and how was calculated the ug/mL concentration?

Response:   

              We firstly made a high concentration of PRE-HIF and used as the stock solution of the extract for treating the cells in this study. 0.1 g PRE-HIF was dissolved in DMSO (final volume:1ml) to make the stock solution of the extract (0.1 mg/mL DMSO). Four µL of stock extract solution was added into 996 µL of new culture medium (the final concentration of PRE-HIF for this condition = 400 µg/mL culture medium), and diluted to each concentration (50 - 400 µg/mL). The concentration of DMSO in control, as presented “0” was equivalent to that in PRE-HIF 400 µg/mL, maximum concentration (DMSO 0.4%). We have added description of preparation of PRE-HIF solution in chapter 2.8.

7) From how much rice was obtained the PRE-HIF extract?

Response: 

              As mentioned above (response 5), 1 kg of purple rice grain yielded about 21.6 g of PRE-HIF. We finally used the purple rice grains more than 20 kg to obtain enough amount of PRE-HIF for all experiment. For quality control of extraction processing, the concentration of anthocyanins in PRE-HIF were compared among batches. As suggested by reviewer 2, we have modified description of the extraction method in chapter 2.1.

8) Chapter 3.5 Pag. 8 row 255. Fig.4 maybe intended as fig. 5?

Response: 

As mentioned by reviewer 2, we have corrected the figure number in chapter 3.5.

9) page 8 Row 259 Avoid potently, some proteins do not seem so downregulated.

Response: 

              We agree with you that “potently” seem to exaggerate, according to the representative band of some proteins. Thus, we have deleted it in chapter 3.5.

10) page 3 row 130 Please add the quantity (volumes and concentrations) of PRE-HIF. Add also the range of concentration used for C3G, or P3G added.

Response: 

              As suggested by reviewer 2, we have added description about the range of concentration in chapter 2.8.

11) Figure 2 E. Phospho ERK ½ does not seem so down regulated

12) Figure 3D. The down regulation of proteins P38MAPK, ERK1/2, phosphor ERC1/2, AMPKalfa, FAS, do not seem so significantly.

Response: 

              We semi-quantified the band density using ImageJ and have added the relative band density below the figure of blotting results to highlight the down-regulation of proteins. Further, relative density score in each group was statistically analyzed. As mentioned by reviewer 2, the down-regulation of pERK1/2 (the ratio of p-ERK/ERK) in both lateral prostates of TRAP rats and PCai1 tumor was not significant. On the other hand, p-p38 MAPK, cyclinD1, cdk 4, p-AMPKα and FAS in the rat prostate tissue and PCai1 xenografts were significantly down-regulated by 1% PRE-HIF. With reflect to these results, we have added these quantitative data in new Figure 3 (TRAP) and new Figure 5 (PCai1 xenograft) and modified description about the expression of ERK1/2 in Abstract and chapter 3.2.

13) Figure 4E and 4F. The downregulation of most proteins is not so strikingly: see p21, AMPKalpha, FAS.

Response: 

              We semi-quantified the band density using ImageJ and have added the relative band density below the figure of blotting results. As suggested by reviewer 2, p21 expression was not obviously down-regulated in LNCaP, as shown in new Figure 6e. The expression of p-AMPK-α/AMPK-α in LNCaP, and FAS/actin in LNCaP and PCai1 exhibits a tendency to down-regulated in the increasing doses of PRE-HIF. With reflect to these results, we have modified description of western blotting in chapter 3.4.

Reviewer 3 Report

Dear Authors,

The manuscript is well written and provides valuable information regarding the effects of isolated anthocyanins from purple rice. Below, I list some questions and comments about the manuscript.

- Section 2.1: The information in this section must be improved. Please, provide the details of the extraction process or indicate a reference, justify the use of selected solves, and provide the results of the HPLC analysis for cyanidin-3-O-glucoside chloride and peonidin-3-O-glucoside chloride.

- Section 2.4: Please justify the selection of PRE-HIF levels in the diet of animals.

- Sections 2.1 and 2.5: It is not clear the reason to carry out two in vivo experiments and do not carry out the comparison between them.

- Section 2.8: It is not clear if the data presented in Table

- Lines 185-189: The lack of apoptotic effect is an important outcome that limits the use of purple rice extract. Please provide an explanation for this outcome in the Discussion section.

- Lines 197-198: Please provide an explanation for the lack of effect on the vessel number after purple rice extract intake in the Discussion section.

- Section 3.5: Please check the number of figures.

- 4. Discussion: This section is too short and must be improved. The effect of anthocyanin-rich extracts targeting anti-cancer effects have been largely studied in the scientific literature. It would of great value to include data from other scientific studies about the effect of C3G chloride, P3G chloride or purple rice against the progression of prostate cancer in order to indicate the role and significance of purple rice among other natural sources. Another important information that must inserted in the discussion is the association or not of in vitro and in vivo data obtained in the study with special attention to purple rice extract.

- Table 4 and 5: Please revise the material and methods section in order to include the range of concentrations used to obtain these results.

Author Response

Comment from reviewer 3: Thank you very much for giving us the chance to revise our manuscript. Your kind comments are useful and help us to improve the quality of our work. The response and modifications are as follows according to your comments.

A list of comments

1) Section 2.1: The information in this section must be improved. Please, provide the details of the extraction process or indicate a reference, justify the use of selected solves, and provide the results of the HPLC analysis for cyanidin-3-O-glucoside chloride and peonidin-3-O-glucoside chloride.

Response:

              Purple rice grain was firstly extracted by using 80% ethanol to obtain the anthocyanin-rich extract from purple rice. The extract was filtered, evaporated, and subsequently freeze-dried to produce crude ethanolic extract (PRE) of the purple rice extract. By using a phase to phase extraction, the parent ethanolic extract was partitioned by hexane, with the two phases then being separated into a hexane soluble fraction (PRE-HSF) and a hexane insoluble fraction (PRE-HIF). Recently, the results from HPLC analysis of the purple rice extract fractions has online published [a new reference #9, as indicated below]. The main anthocyanins, cyanidin-3-glucoside (C3G) and peonidin-3-glucoside (P3G) in PRE-HIF was 4.87 ± 0.05 and 2.25 ± 0.02 mg/g extract, respectively. However, both anthocyanins was not found in PRE-HSF, which contains high amount of vitamin D derivatives and gamma-oryzanol. These results indicate that hexane could effectively isolate lipophilic compounds from the ethanolic extract, so that anthocyanins were mainly presented in the residue of fractionation (PRE-HIF). As suggested by reviewer 3, we have added a referencer and descriptions in chapter 2.1. Moreover, we also have revised the discussion for justifying the use of selected solvent in the present study.

Reference

Yeewa, R.; Sakuludomkan, W.; Kiriya, C.; Khanaree, C.; Chewonarin, T. Attenuation of benign prostatic hyperplasia by hydrophilic active compounds from pigmented rice in testosterone implanted rat model. Food & Function 2020, https://doi.org/10.1039/C9FO02820J.

2) Section 2.4: Please justify the selection of PRE-HIF levels in the diet of animals.

Response: 

              We calculated %PRE-HIF in the diet as accordance with carbohydrate intake of human. Recommended dietary allowance (RDA) for selected nutrients has reported that healthy 25-50 years-old adults who weight 70 kg get carbohydrate approximately 364 g/day (5.2 g/kg.bw/day) [Meisenberg, G.; Simmons, W.H. Principles of Medical Biochemistry E-Book; Elsevier Health Sciences: 2016.]. Our recent study has reported 5.2 g of purple rice grain yields 0.11 g of PRE-HIF (2.16%), indicating that daily intake of PRE-HIF is a 0.11 g (/kg/day). It was similar amount to the intake in low-dose PRE-HIF (0.2%) group of TRAP rat in the present study as shown in Table 1. We also used a high-dose (1%) group to observe dose-dependent effects and consider difference of metabolizing enzyme activity between human and animals.

The intake of PRE-HIF in TRAP rats was also similar range to that in the benign prostatic hyperplasia (BPH) model previously [a new reference #9]. We reported that oral administration of PRE-HIF 0.1 and 1 g/kg/day could retard prostate enlargement and improve histological changes in the rats’ prostate tissues, while no signs or symptoms of toxicity were found in any of BPH rats. Like BPH, the progression of prostate cancer is initially associated with androgen receptor (AR) signaling pathway. Therefore, we hypothesized that PRE-HIF may inhibit prostate carcinogenesis. As suggested by reviewer 3, we have added average PRE-HIF intake for rats in chapter 3.1 and modified Table 1, and data for mice in chapter 3.3.

3) Sections 2.1 and 2.5: It is not clear the reason to carry out two in vivo experiments and do not carry out the comparison between them.

Response: 

              Purple rice has been reported to protect various types of cancer, whereas its potential action on prostate carcinogenesis and castration-resistant prostate cancer (CRPC) have not yet been elucidated. In the present study, effects of PRE-HIF, an anthocyanin-rich fraction from purple rice extract, on prostate carcinogenesis were investigated in vivo by using TRAP model (chapter 2.4). Furthermore, the effects of PRE-HIF against CRPC were explored in rat CRPC xenograft model of PCai1 cells (chapter 2.5). The results from the present study provides the rationale for the use of purple rice as a dietary supplement in the prevention or treatment of prostate cancer. As suggested by reviewer 3, we have added descriptions about the common effects between 2 experiments in Discussion.

4) Section 2.8: It is not clear if the data presented in Table

Response: 

              Thank you very much for your kind comment. We feel sorry about the unclear statement. To better explain the cell proliferation assay, we have revised chapter 2.8.

5) Lines 185-189: The lack of apoptotic effect is an important outcome that limits the use of purple rice extract. Please provide an explanation for this outcome in the Discussion section.

Response: 

              Generally, anti-apoptotic effect is one of the main mechanisms underlying the potential protective effects of natural or synthetic compounds against various types of cancers. In this study, 1% PRE-HIF reduced Ki-67 index by more than 20% (from 79.7 ± 16.8% to 54.3 ± 18.8%) in lateral prostate of TRAP model, while it did not alter apoptotic index (control: 3.8 ± 1.0%, 1% PRE-HIF: 3.3 ± 1.4%, respectively). As accordance with previous studies, TUNEL indices were increased by around 5 - 6%, at least less than 10% with induction of apoptosis by test chemicals in prostate cancer tissue [13,17]. This indicates that a strong anti-proliferative effect may provide chemopreventive or chemotherapeutic effects without induction of apoptosis. Therefore, we believe that the lack of apoptotic effect would not attribute to the limits in the use of purple rice extract. With reflect to the comment by reviewer 3, we have added descriptions in Discussion.

6) Lines 197-198: Please provide an explanation for the lack of effect on the vessel number after purple rice extract intake in the Discussion section.

Response: 

              Anti-angiogenesis is one of the underlying mechanisms that can block the tumor growth. The present study suggests that PRE-HIF suppressed CRPC tumor growth via decrease of cell proliferation in PCai1 tumor, even though no effect of on the vessel number was observed. PRE-HIF may suppress the cancer invasion and metastasis by targeting to other mechanisms, such as inhibition of cancer cell migration, because its main component is anthocyanin, though further studies are necessary to conclude [a new reference #30, as indicated below]. As recommended by reviewer 3, we have added descriptions about angiogenesis in Discussion.  

Reference

Zhou, J.; Zhu, Y.-F.; Chen, X.-Y.; Han, B.; Li, F.; Chen, J.-Y.; Peng, X.-L.; Luo, L.-P.; Chen, W.; Yu, X.-P. Black rice-derived anthocyanins inhibit HER-2-positive breast cancer epithelial-mesenchymal transition-mediated metastasis in vitro by suppressing FAK signaling. International journal of molecular medicine 2017, 40, 1649-1656, doi: 10.3892/ijmm.2017.3183..

7) Section 3.5: Please check the number of figures.

Response: 

As mentioned by reviewer 2, we have corrected the figure number in chapter 3.5.

8) 4. Discussion: This section is too short and must be improved. The effect of anthocyanin-rich extracts targeting anti-cancer effects have been largely studied in the scientific literature. It would of great value to include data from other scientific studies about the effect of C3G chloride, P3G chloride or purple rice against the progression of prostate cancer in order to indicate the role and significance of purple rice among other natural sources. Another important information that must inserted in the discussion is the association or not of in vitro and in vivo data obtained in the study with special attention to purple rice extract.

Response: 

              As suggested by reviewer 3, we have added description about anti-cancer effect of anthocyanins and references in Discussion. We confirmed anti-proliferative effect of PRE-HIF on prostate cancer in both in vitro and in vivo, and detected the common protein expression changes. Therefore, we believe that their results are associated with together. The description was modified in Discussion.

References

Tanaka, J.; Nakamura, S.; Tsuruma, K.; Shimazawa, M.; Shimoda, H.; Hara, H. Purple rice (Oryza sativa L.) extract and its constituents inhibit VEGF-induced angiogenesis. Phytother Res 2012, 26, 214-222, doi:10.1002/ptr.3533. Ding, M.; Feng, R.; Wang, S.Y.; Bowman, L.; Lu, Y.; Qian, Y.; Castranova, V.; Jiang, B.-H.; Shi, X. Cyanidin-3-glucoside, a natural product derived from blackberry, exhibits chemopreventive and chemotherapeutic activity. Journal of Biological Chemistry 2006, 281, 17359-17368, doi:10.1074/jbc.M600861200. Chen, Y.F.; Shibu, M.A.; Fan, M.J.; Chen, M.C.; Viswanadha, V.P.; Lin, Y.L.; Lai, C.H.; Lin, K.H.; Ho, T.J.; Kuo, W.W., et al. Purple rice anthocyanin extract protects cardiac function in STZ-induced diabetes rat hearts by inhibiting cardiac hypertrophy and fibrosis. J Nutr Biochem 2016, 31, 98-105, doi:10.1016/j.jnutbio.2015.12.020. Antonarakis, E.S.; Lu, C.; Luber, B.; Wang,H; Chen, Y.; Zhu, Y.; Silberstein, J.L.; Taylor, M.N.; Maughan, B.L.; Denmeade, S.R., et al. Clinical Significance of Androgen Receptor Splice Variant-7 mRNA Detection in Circulating Tumor Cells of Men With Metastatic Castration-Resistant Prostate Cancer Treated With First- and Second-Line Abiraterone and Enzalutamide. J Clin Oncol. 2017, 35, 2149-2156. doi: 10.1200/JCO.2016.70.1961. Yang, W.-H.; Xu, J.; Mu, J.-B.; Xie, J. Revision of the concept of anti-angiogenesis and its applications in tumor treatment. Chronic diseases and translational medicine 2017, 3, 33-40, doi:10.1016/j.cdtm.2017.01.002. Lu, J.; Zhang, K.; Nam, S.; Anderson, R.A.; Jove, R.; Wen, W. Novel angiogenesis inhibitory activity in cinnamon extract blocks VEGFR2 kinase and downstream signaling. Carcinogenesis 2010, 31, 481-488, doi: 10.1093/carcin/bgp292. Mapoung, S.; Suzuki, S.; Fuji, S.; Naiki‐Ito, A.; Kato, H.; Yodkeeree, S.; Ovatlarnporn, C.; Takahashi, S.; Limtrakul, P. Cyclohexanone curcumin analogs inhibit the progression of castration‐resistant prostate cancer in vitro and in vivo. Cancer science 2019, 110, 596-607, doi:10.1111/cas.13897. Zhou, J.; Zhu, Y.-F.; Chen, X.-Y.; Han, B.; Li, F.; Chen, J.-Y.; Peng, X.-L.; Luo, L.-P.; Chen, W.; Yu, X.-P. Black rice-derived anthocyanins inhibit HER-2-positive breast cancer epithelial-mesenchymal transition-mediated metastasis in vitro by suppressing FAK signaling. International journal of molecular medicine 2017, 40, 1649-1656, doi:10.3892/ijmm.2017.3183. Chandrasekar, T.; Yang, J.C.; Gao, A.C.; Evans, C.P. Mechanisms of resistance in castration-resistant prostate cancer (CRPC). Transl Androl Urol 2015, 4, 365-380, doi:10.3978/j.issn.2223-4683.2015.05.02.

9) Table 4 and 5: Please revise the material and methods section in order to include the range of concentrations used to obtain these results.

Response: 

              As mentioned by reviewer 3, we realized that we should revised the chapter 2.8 in order to show the range of concentrations used to obtain the results in Figure 4 and 5. Therefore, we have corrected chapter 2.8 as shown in the revised manuscript.

Round 2

Reviewer 1 Report

In response to my previous concerns, this manuscript has indeed improved substantially. However, I hope that the author will check the data in Table 2 again. Is the SE (standard error) actually the standard deviation (SD)?

Reviewer 3 Report

Dear Authors,

The revised version of the manuscript is adequate.